*The Company of*
**Biologists**

# Computer vision analysis to identify episodic flapping in hovering hummingbirds

Maria Ximena Bastidas-Rodriguez[1,2], Ana Melisa Fernandes[1], María José Espejo Uribe[1], Diana Abaunza[1], Ashley Smiley[3], Juan Sebastián Roncancio[4], Eduardo Aquiles Gutierrez Zamora[5], Cristian Flóres Pai[6], Kristiina Hurme[1,7], Christopher J. Clark[8] and Alejandro Rico-Guevara[7,9,*]

## ABSTRACT

Hummingbirds hover, enabling them to serve as exclusive pollinators for many plant groups with which they have coevolved. They combine bird muscle power with insect flight skills to hover and maneuver in any direction. This study investigates an under-explored aspect of their hovering flight: many species exhibit an alternative gait, in which they flap their wings discontinuously, including momentary pauses sporadically interspersed between short series of wingbeats, which collectively generate a pattern we denote as 'intermittent hovering'. These brief pauses have not been characterized in hummingbirds but may play aerodynamic, energetic, and/or signaling roles. To detect and measure intermittent hovering, we present a high-throughput method to quantify these irregular wingbeat pauses using computer vision and signal analysis. High-speed videos of 11 species collected by recording free-living hummingbirds allowed us to track and analyze the pauses during hovering. The proposed algorithm has a precision of 74%, accurately detecting flapping pauses and thus intermittent hovering. Most of the misclassification errors were false positives: when very still hovering hummingbirds were continuously moving their wings, but some of the consecutive frames were similar enough that the algorithm classified them as a brief pause. These false positives, however, are easily discarded upon quick visual inspection, and the algorithm had only 19 false negatives across all videos, actually detecting intermittent hovering if it was present. This method is a step forward in creating tools that help researchers analyze complex behavioral patterns. Our study confirms the feasibility of reliably detecting intermittent hovering, contributing to our understanding of hummingbird flight dynamics.

KEY WORDS: Computer vision, Deep learning, Hummingbirds, Pausing behavior, Zero learning

[1]Centro de investigación Colibrí Gorriazul, Fusagasugá, Cundinamarca, Colombia. [2]Parquesoft Nariño, Pasto, Nariño, Colombia. [3]Department of Integrative Biology, University of California, Berkeley, CA 94720, USA. [4]Ministerio de Minas y Energía, Bogotá D.C. 111061, Colombia. [5]Departamento de Biología Universidad de Nariño, Laboratorio de Palinología e Interacción Planta Animal, Pasto, Nariño 52001, Colombia. [6]Fundación Ecológica Los Colibríes de Altaquer (FELCA), Altaquer, Nariño, Colombia. [7]Department of Biology, University of Washington, Seattle, WA 98105, USA. [8]Department of Evolution, Ecology, and Organismal Biology, University of California, Riverside, CA 92521, USA. [9]Burke Museum of Natural History and Culture, University of Washington, Seattle, WA 98105, USA.

*Author for correspondence (colibri@uw.edu)

M.X.B.-R., 0009-0007-3087-5456; A.R.-G., 0000-0003-4067-5312

## INTRODUCTION

Understanding the mechanics of animal locomotion is essential for elucidating the intricate interplay of behavioral goals, energetics, and biomechanical constraints. In terrestrial locomotion, changes in speed trigger distinct kinematic shifts, manifesting as specific gaits – such as the transition from walking to jogging in humans (Minetti, 1998) and the varied gaits of horses, including walking, trotting, cantering, and galloping (Hoyt and Taylor, 1981). These gait transitions are not confined to land (Alexander, 2003), they also occur in aquatic (Videler and Nolet, 1990) and aerial environments (Rayner, 1988; Tobalske, 2007). The mechanics of these gaits reflect complex relationships that balance energy conservation with locomotor dynamics (Alexander, 2003). Notably, gait shifts are primarily influenced by two main factors: (1) the minimization of travel costs at specific speeds (Alexander, 2003) and (2) the recruitment of different muscle groups or the navigation of biomechanical and physiological constraints to achieve the desired speed (Hale et al., 2006).

While early research on flying animals suggested a dichotomous approach to flight patterns – separating steady flight into distinct categories such as continuous flapping and flapping pauses (Rayner, 1988) – subsequent investigations indicate that these transitions are often more gradual and nuanced (Tobalske et al., 2007). Intermittent flight, characterized by episodic flapping pauses, involves a modification of the continuous wingbeat pattern that occurs at irregular intervals rather than in a continuous rhythm (Tobalske, 2010). This flight mode allows animals to alternate between flapping and pausing (gliding/bounding) (Rayner, 1985), where energy conservation is one hypothesis to explain the occurrence of this behavior (Tobalske, 2016a,b). Although much of the literature on flight focuses on forward movement, here, we document a similar transition that occurs in stationary flight in hummingbirds, termed intermittent static hovering (Hurme et al., 2026 preprint). Hummingbirds' unique flight mechanics allow them to sustain hovering for more than a few seconds by generating lift during both the downstroke and, to a lesser extent, the upstroke due to highly flexible shoulders (Warrick et al., 2005), maintaining wing rotation throughout the stroke cycle (Tobalske, 2016a,b). Here, we present a methodology to easily quantify the occurrence of episodic pauses during hovering flight, testing a variety of hummingbird species. The lack of research on hummingbirds' intermittent hovering highlights a significant gap in our understanding of their specialized flight behavior and the potential reasons for its evolution.

Hummingbirds are thought to have evolved sustained hovering to facilitate visiting delicate pendant flowers with elongated tubular shapes and narrow entrances (Warrick et al., 2012). Morphological and physiological adaptations, such as specialized wing shapes, further facilitate their unique hovering capabilities and high wingbeat frequencies (Altshuler and Dudley, 2002; Greenewalt, 1962). With over 360 recognized species (Winkler et al., 2020),

hummingbirds exhibit remarkable diversity. Many species can coexist and interact in a given area, for example, around the Río Ñambí area, in Nariño, Colombia, where at least 29 species coexist (Hilty and Brown, 1986; Gutiérrez et al., 2004). Many species have evolved diverse competitive strategies, ranging from niche partitioning, associated with bill-corolla matching and exploitative competition, to intense resource defense, where aggressive displays and interference competition determine resource access, often involving both intraspecific and interspecific competition, as many species can feed from certain flowers (Rico-Guevara et al., 2021; Sargent et al., 2021; Tellez-Colmenares, 2018).

The potential aerodynamic, energetic, and/or signaling associations of this behavior will be unveiled only by future study and prolific data collection. The majority of the work on hummingbird flight biomechanics has focused on a single hummingbird clade, the 'bee' hummingbirds (Mellisugini), which constitute the most common hummingbirds found in the USA and Canada and thus are the ones that are the most accessible to flight laboratory facilities. On the other hand, the majority of the species occur outside those countries, and outside the bee clade, many exhibit puzzling and unstudied wing traits (e.g. underwing coloration, Ayerbe-Quiñones, 2015; Jiménez and Ornelas, 2015) and kinematics. For instance, Mobbs (1982) observed in coronets (genus *Boissonneaua*) that: "A characteristic of this genus is the unusual wing action which is used almost every time these birds alight … [in which] the wings will be held almost vertical for a second before being folded into the resting position.". Although there is recent research on wing kinematics for tropical hummingbirds (e.g. Ingersoll et al., 2018; Díaz-Salazar et al., 2024), significant gaps remain in our knowledge about most of the species, and efficient methods of high-speed video analysis could facilitate the study of this large number of understudied taxa.

Traditional high-speed video techniques for studying hummingbird flight patterns often entail labor-intensive frame-by-frame analysis. However, advancements in computer vision and video processing are paving the way for more efficient methods of studying these behaviors. For instance, Serrano et al. (2018) employed deep learning to track hummingbirds in video frames for nest monitoring, while Weinstein (2018) developed a movement detection system that contrasts each frame with a static background image. Martínez et al. (2015) analyzed wing motion through multiscale dense optical flow, yielding global measures of angular acceleration. In a recent study (Bastidas-Rodriguez et al., 2024 preprint), we introduced a method for estimating wingbeat frequencies in 11 hummingbird species via video analysis and computer vision, allowing for efficient detection, classification, and tracking without manual labeling. This method used detected bounding boxes to calculate the intersection over union (IoU) metric across consecutive frames. This metric generated signals that allowed for short-term Fourier transform analysis to monitor frequencies over time.

Here, we study an intriguing hummingbird flight behavior: the ability to pause their wings momentarily at the end of the upstroke and continue to hover (Hurme et al., 2026 preprint). This midair wing pausing interrupts the continuous wing motion at the point in which the wings are extended backwards, aligning with a typical position during mid-pronation that bridges the wing movements between the upstroke and downstroke (Fig. 1). This research presents a novel computer-based method to characterize this intermittent static hovering behavior. We compared automated assessments with visual inspections conducted by three trained observers. Initially, two observers conducted the assessments, followed by a review and consensus check by a third observer, specifically focused on detecting wing motion pauses.

## MATERIALS AND METHODS

Our proposed method (Fig. 2) builds on the modules from Bastidas-Rodriguez et al. (2024) for detection, classification, and tracking, while introducing a novel module specifically designed for monitoring pauses in flight behavior. The system is composed of four key components:

- **Detection module:** This component utilizes the YOLO-NAS (You Only Look Once Neural Architecture Search) algorithm to accurately identify and localize objects within bounding boxes, ensuring reliable initial detection of subjects in the video frames.
- **Reclassification method:** We implement the Open Clip (Patil, 2024; Radford et al., 2021) algorithm, enhanced by a text prompt, to refine classifications for relevant categories, such as hummingbirds and flowers. This step allows for improved specificity in identifying objects of interest.
- **Tracking system:** Our tracking component compares image features extracted by Open Clip across sequential frames. This enables continuous monitoring of identified subjects, enhancing the robustness of our analysis.
- **Pausing behavior module:** This new module employs Jaccard's coefficient to quantify movement by analyzing the detection boxes across video frames. This metric allows us to determine instances of hovering pauses accurately.

We provide a detailed description of each component of our methodology, including the computational and data resources employed for evaluation. Our approach closely adheres to the methods outlined in our paper (Bastidas-Rodriguez et al., 2024 preprint), ensuring consistency and reliability in our analysis.

### Dataset and computational resources

We recorded high-speed videos of 11 species of wild hummingbirds (Table S1) performing static hovering at hummingbird feeders illuminated by sunlight and with a natural background (far vegetation to not interfere with the identification of the bird) and visually measured their wing flapping frequency (Steen, 2014). The recordings were made using fixed-point cameras, specifically a Fastec

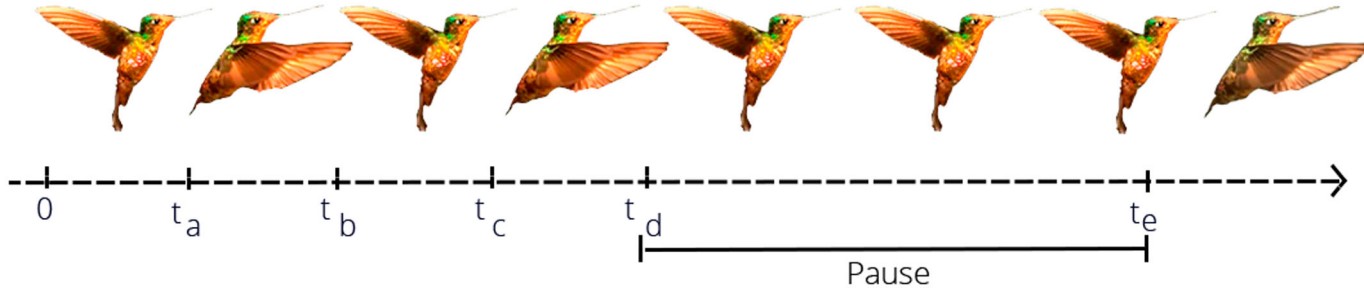

**Fig. 1. Timeline of hummingbird flight states.** Timeline of the states during the hummingbird flight. During the intervals [0, $t_a$] and [$t_b$, $t_c$], the bird is in an upstroke state. During the intervals [$t_a$, $t_\beta$] and [$t_c$, $t_d$], the bird is in a downstroke state. In the interval [$t_d$, $t_e$], a pause behavior is observed, where the bird temporarily suspends wing movement.

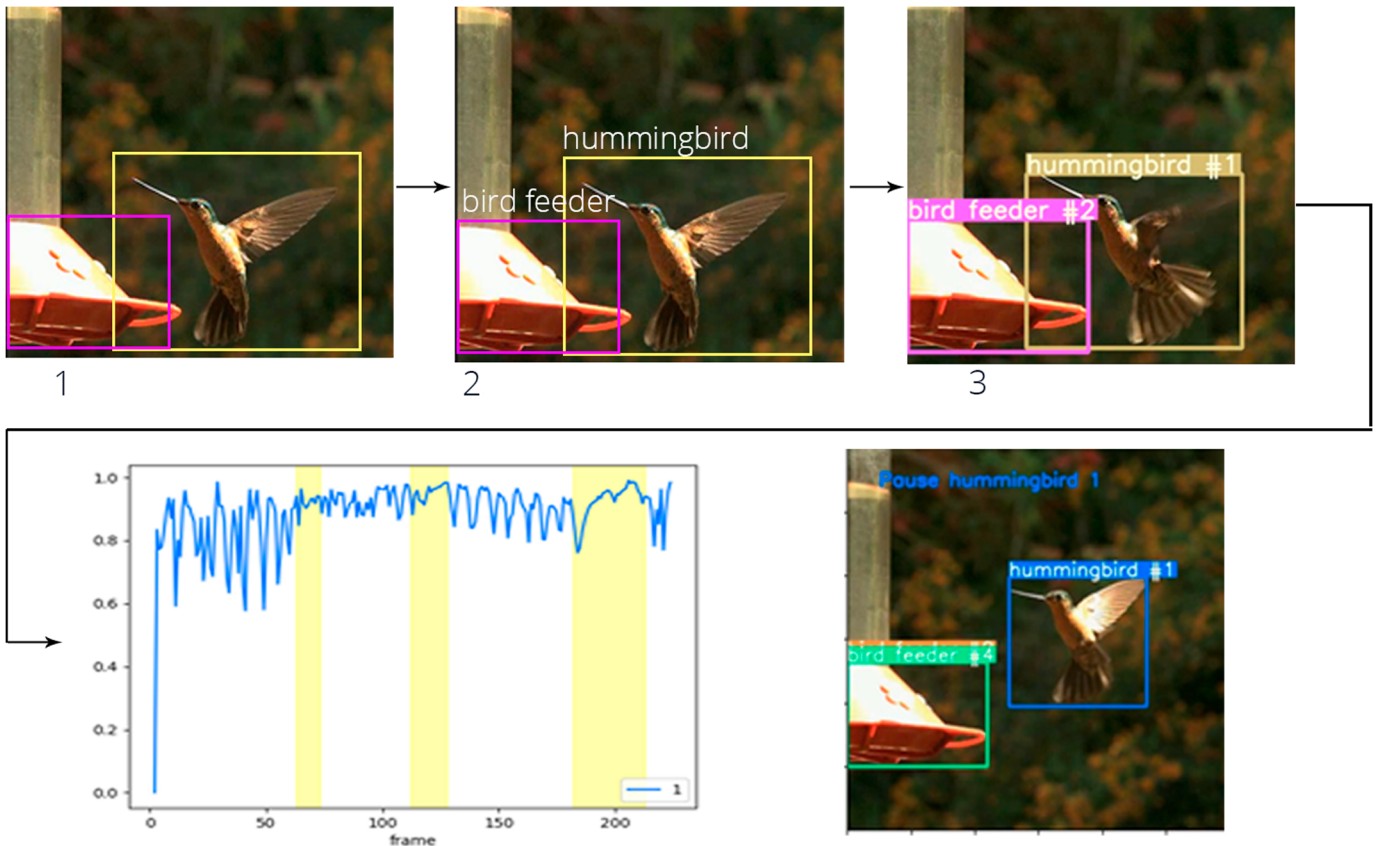

**Fig. 2. Steps of the proposed algorithm.** (1) Object detection stage: Utilizes the YOLO-NAS algorithm to identify and localize objects within the scene. (2) Classification stage: Employs a text prompt-based approach with OpenAI's CLIP model to categorize detected objects. (3) Object tracking stage: Tracks identified objects across frames to maintain their identity and position over time. (4) Pausing detection stage: Implements a custom method to detect and analyze instances of pausing behavior in hummingbirds.

HS7, a Fastec TS5, or a Chronos 2.1 camera, all operating at a minimum resolution of 800×600 pixels and capturing footage at frame rates of 500 frames per second. The wingbeat frequency of these species was found to be approximately 35 to 45 Hz (Bastidas-Rodriguez et al., 2024 preprint), meaning each wingbeat was captured in about 14 video frames. Videos recorded at higher speeds could augment the number of frames where a pause/wingbeat is recorded; this information could be very useful for the algorithm, as a pause could be easily identified. In turn, this would reduce FN as a pause would last for longer periods of time than other movements, and it would reduce FP as the image quality given by the movement would be smoother, resulting in sharper images on each frame, particularly for the fastmoving wings. Furthermore, each frame's RGB color space was analyzed individually through our proposed processing pipeline. Although illumination conditions and camera settings were not standardized, we selected video sequences where the hummingbird was the primary focus of the scene, ensuring minimal backlighting and consistent lighting conditions throughout the video.

In total, 13 videos were chosen for analysis to compare the performance of our proposed algorithm against the visual estimation method. Each video contained only one hummingbird in flight, although other hummingbirds could be present statically (perched) at the feeders. This selection criterion allowed for focused analysis on a single subject as the main object in each scene. Additionally, we identified instances of flight pauses within the videos. The proposed method would serve as a pilot validation of the pausing behavior detection approach.

### Detection, classification, and tracking
In this study, we adopted the methodology outlined by Bastidas-Rodriguez et al. (2024). For object detection, we utilized the YOLO architecture (Jiang et al., 2022), a widely recognized deep neural network renowned for its efficiency in real-time object detection. This algorithm was employed to extract

bounding boxes around regions containing hummingbirds. Subsequently, these bounding boxes were classified using the multimodal model CLIP (contrastive language-image pre-training) (Radford et al., 2021), which effectively compares text and image features in a contrastive manner, facilitating the accurate identification and differentiation of detected objects.

Following the classification process, we compared the extracted image features across frames to track the hummingbird's movement. In our previous study, we achieved F1-scores of 0.76 for detection and 0.91 for classification, with a score of 1.0 indicating optimal performance. Given that our approach did not involve manual labeling, we consider these results to be satisfactory.

### Detection of flying pausing behavior
We propose a method for detecting pauses in hummingbird flight by analyzing the bounding boxes of tracked individuals. Specifically, we measure the Jaccard (1912) coefficient (Eqn 1) between each pair of sequential frame detections (Fig. 3). This coefficient quantifies the ratio of the intersection to the union of the bounding boxes from consecutive frames:

$$J\left(area_n, area_{n+1}\right) = \frac{|area_n \cap area_{n+1}|}{|area_n \cup area_{n+1}|}. \quad (1)$$

In this equation, $area_n$ represents the area of the bounding box for $hummingbird_x$ in $frame_k$, while $area_{n+1}$ denotes the area of the bounding box for the same $hummingbird_x$ in $frame_{k+1}$. By characterizing the Jaccard coefficient over time, we can analyze the movement patterns of the hummingbird throughout the video duration.

To detect pauses, we examine the derivative of the Jaccard coefficient curve (Eqn 2). A derivative value approaching zero indicates minimal change between frames. When these changes fall below a specified threshold – set to 90% of the mean Jaccard coefficient (designated as $pause_{threshold}$) – we infer a likelihood of detecting a pause at that

moment. This analysis enables us to identify instances of stationary behavior, providing valuable insights into the flight dynamics of hummingbirds.

$$Pause = \{1 \; if \; 0 \; <= \frac{d\,J(area_n, \; area_{n+1})}{dt} <= mean(dJ/dt) \times pause_{threshold}\} \quad (2)$$

### Evaluation metrics

We employed three standard metrics from the field of computer vision to evaluate our model's performance: precision, recall, and F1-score. These metrics are calculated based on the values of TPs, FPs, and FNs.

**Precision:** This metric measures the proportion of TPs among all positive predictions made by the model, providing insight into the accuracy of the positive classifications.

$$Precision \; = \; \frac{TP}{TP \; + \; FP} \quad (3)$$

**Recall:** This metric reflects the proportion of TPs among all actual positives (groundtruth), indicating the model's ability to identify relevant instances.

$$Recall \; = \; \frac{TP}{TP \; + \; FN} \quad (4)$$

**F1-score:** The F1-score combines precision and recall by calculating their harmonic mean, providing a balanced measure of the model's performance. The best possible value for the F1-score is 1.0, indicating perfect precision and recall.

$$F1\text{-score} = \frac{2Precision \; \times \; Recall}{Precision \; + \; Recall} \quad (5)$$

## RESULTS

In total, we analyzed 13 videos and birds of 11 species (one species with three different individuals/videos) of four different clades with one or more pauses per video (Table S1). We identified the sex of nine species, one of which we were able to capture videos for both sexes (Table S1). We identified 55 true positives (TPs) where a pause was correctly predicted, 19 false positives (FPs) where a pause was predicted but not labeled as such, and 19 false negatives (FNs) where a pause was present but not detected by our method (Table S1), resulting in a precision of 74%, a recall of 74%, and an F1-score of 0.74. These metrics demonstrate the effectiveness of our method in accurately detecting pauses. The visual representation in Fig. 4 offers a comprehensive overview of these outcomes, showcasing examples of frames where predicted pauses align with manual annotation of pauses, indicating a high degree of accuracy in our detection mechanism.

*Saucerottia saucerottei* exhibits the fewest pauses relative to the number of wingbeats, indicating that pauses are infrequent in this species (Fig. 5). In contrast, *Pterophanes cyanopterus* shows a higher mean duration of pauses (see manual value in Fig. 5) and the highest number of pauses per wingbeat, with more than half of its wingbeats (27 pauses out of 48 wingbeats) accompanied by a pause

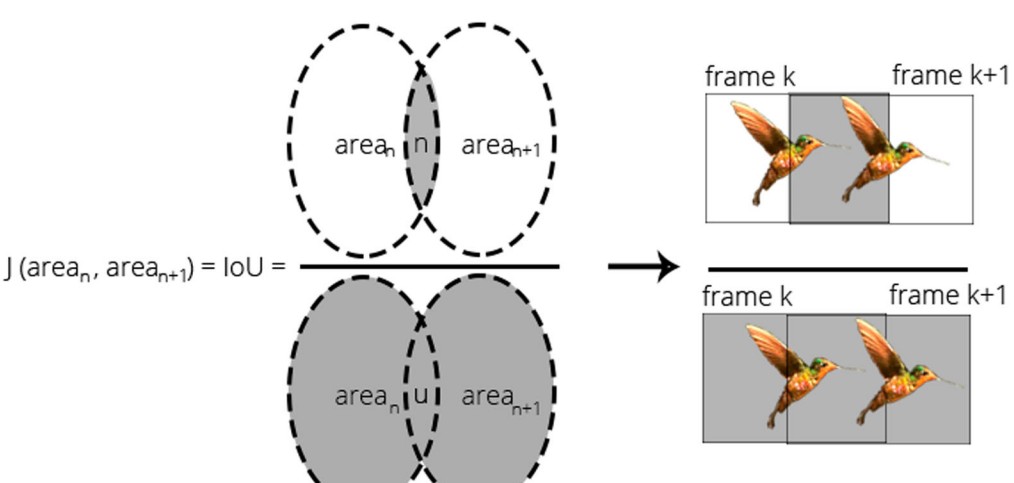

**Fig. 3. Jaccard's coefficient representation.** Shows the intersection over the union of the bounding boxes of a tracked hummingbird between consecutive frames. The plot displays Jaccard's coefficient versus frame number, with plateaus indicated by yellow vertical lines, highlighting regions where a pause was predicted. These plateaus suggest minimal movement, corresponding to potential pauses in the hummingbird's flight.

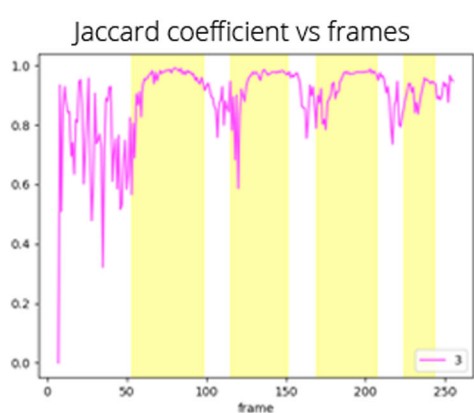

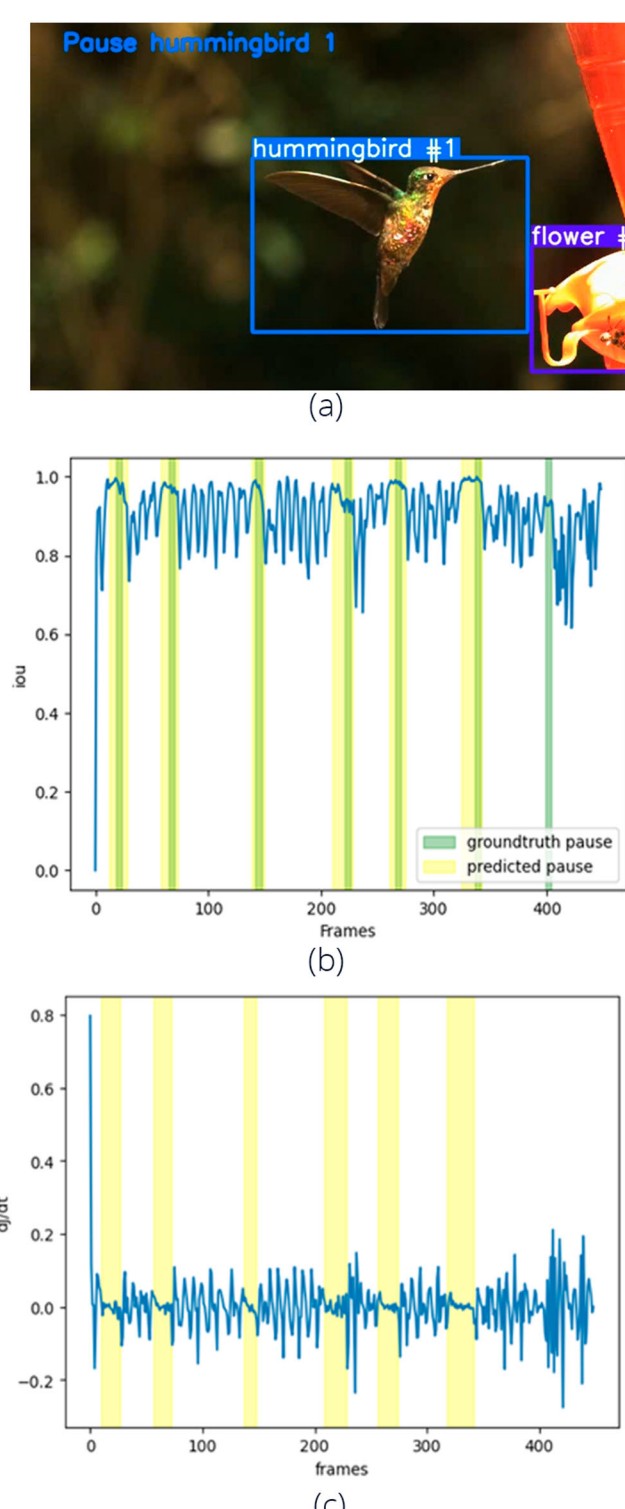

**Fig. 4. Visual analysis of the proposed pausing identification method.** (a) Displays a frame from one of the videos where a pause was detected. (b) Shows the Jaccard coefficient between consecutive frames over time, with yellow lines indicating the pauses detected by the algorithm and green lines representing the pauses labeled by the expert biologist. (c) Presents the derivative of the Jaccard coefficient curve, highlighting changes in motion and identifying stationary periods.

approximately every two wingbeats (Table S2). On the other hand, the algorithm was not able to correctly measure the mean duration of the pauses for most of the species (Fig. 5).

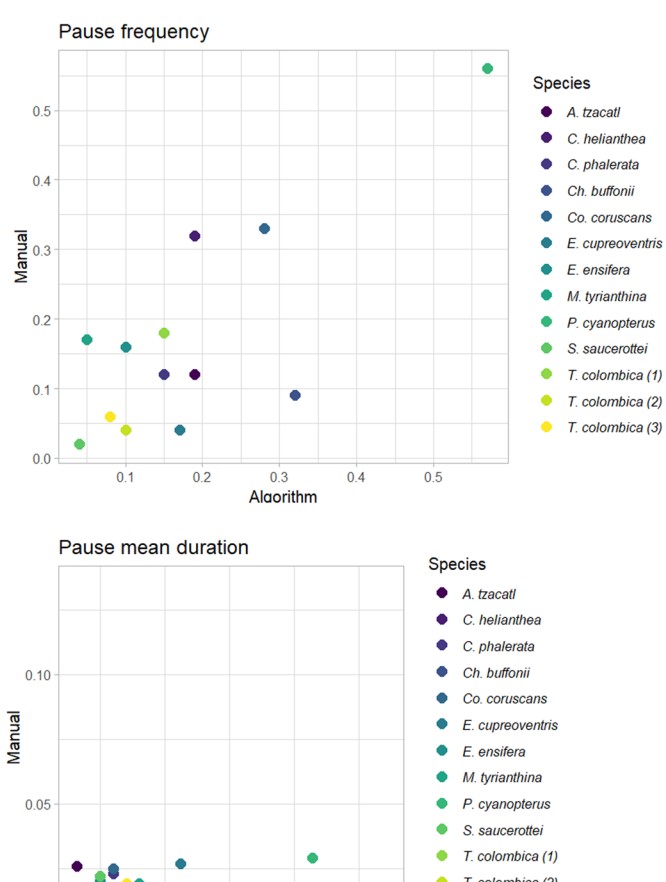

**Fig. 5. Pause frequency (number of pauses/number of wingbeats) and pause mean duration by body mass (g) for 13 individuals of 11 species of hummingbirds (Ayerbe-Quiñones, 2024).** Filled circles denote pause frequency measured by the algorithm and filled triangles wingbeat frequency measured manually.

## DISCUSSION

The pausing algorithm demonstrated its effectiveness in detecting pauses, achieving an F1-score of 0.74. A deeper analysis of the FPs revealed that most errors occurred when the hummingbird was performing hovering flight instead of pausing, maintaining nearly the same position in consecutive frames and capturing wing rotation at the end of the stroke. These subtle movements during hovering were mistakenly classified as pauses by the algorithm. However, the overall number of FPs was relatively low, and not all stationary flight instances were incorrectly identified. Although the algorithm missed some pauses, it performed more accurately when the subject occupied a significant portion of the frame and when the video was captured from a profile angle (Fig. 4). Future improvements could include exploring adaptive thresholding techniques to better accommodate variations in flight behavior, as well as retraining the algorithm to be adapted to different scene angles.

Intermittent hovering, a flight strategy characterized by alternating periods of active wingbeats and brief pauses, presents a discontinuous hovering pattern. One hypothesis for its existence is that intermittent hovering serves as an energy-conservation strategy (Rayner, 1985; Tobalske, 2010). Hovering is known to be one of the most energetically demanding flight modes (Tobalske, 2010) because the induced power output required to hover is high (Rayner, 1999). Another nonexclusive hypothesis is that intermittent hovering

may enhance visual processing during foraging by allowing hummingbirds to take in sensory information during brief pauses, helping them adjust their positioning with greater precision (Altshuler and Dudley, 2002; Ros and Biewener, 2016; Goller et al., 2019).

Another hypothesis invokes biomechanical constraints related to power output modulation and physiological factors like neural control (Tobalske, 2007). The nervous system must coordinate complex motor patterns to ensure precise wing movements during different phases of flight (Chai et al., 1996). Intermittent flight may allow for adaptive changes in neural signaling, enabling birds to respond dynamically to environmental conditions, such as varying wind currents, obstacles in their path, predation risks, and the need for efficient foraging, all requiring birds to modulate their flight behavior. This adaptability could involve enhanced integration of somatosensory information from their wings and body, allowing for quicker adjustments in flight posture and strategy. Additionally, the ability to adjust wingbeat frequency and amplitude during these pauses can enhance stability, further supporting efficient foraging and navigation (Tobalske et al., 2007). A related hypothesis is that because birds have a limited number of muscle fibers in their pectoralis, alternate gaits might help them modulate power output (Rayner, 1999; Tobalske, 2010). By incorporating pauses or glides, birds can reduce the metabolic energy of flight, effectively balancing the need for propulsion with energy conservation (Tobalske et al., 2003). The last hypothesis is signaling; some hummingbirds like *P. cyanopterus* have a striking blue coloration under the wing, so pausing behavior can be used as dominance signaling, showing that color under the wing and/or their ability to perform more frequent and longer midair pauses. Additional hypotheses around intermittent hovering are worth exploring in future work, for instance, its relationship with muscle fatigue or possible neuromuscular constraints (Mahalingam and Welch, 2013; Tobalske, 2010).

The methods proposed in this paper describe a feasible avenue for reliably detecting intermittent hovering, which could expedite data collection for many species, and thus, it expands our potential to understand hummingbird flight dynamics across the family and to explore evolutionary reasons for flight behaviors more generally. Further research could use more advanced methods in the field of computer vision, such as landmark detection (Lin et al., 2022), to overcome the issues faced here. These methods will probably benefit from using better-resolution videos to capture the whole wing movement (e.g. Díaz-Salazar et al., 2024).

## Competing interests
The authors declare no competing or financial interests.

## Author contributions
Conceptualization: M.X.B.-R., A.R.-G.; Data curation: M.X.B.-R., A.M.F., M.J.E.U., D.A., J.S.R., E.A.G.Z., C.F.P., A.R.G.; Formal analysis: M.X.B.-R., A.M.F., M.J.E.U., D.A., A.R.G.; Investigation: M.X.B.-R., A.M.F., A.R.-G.; Methodology: M.X.B.-R., A.M.F., A.R.-G.; Project administration: A.R.-G.; Resources: A.R.-G.; Software: M.X.B.-R.; Writing – original draft: M.X.B.-R., A.M.F., A.R.-G.; Writing – review & editing: M.X.B.-R., A.M.F., J.S.R., E.A.G.Z., A.S., K.H., C.J.C., A.R.-G.

## Funding
This work was supported in part by the Fundación Ceiba, doctoral studies program at National University of Colombia, and the Walt Halperin Professorship provided by the Washington Research Foundation. Open Access funding provided by University of Washington. Deposited in PMC for immediate release.

## Data and resource availability
All relevant data and details of resources can be found within the article and its supplementary information.

## Peer review history
The peer review history is available online at https://journals.biologists.com/bio/lookup/doi/10.1242/bio.061936.reviewer-comments.pdf.

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
