## [Peer Review File · Biology Open]

Computer vision analysis to identify episodic flapping pauses in hovering hummingbirds

Maria Ximena Bastidas-Rodriguez, Ana Melisa Fernandes, María José Espejo-Uribe, Diana Abaunza, Juan Sebastián Roncancio, Eduardo Aquiles Gutierrez-Zamora, Cristian Flórez-Pai, Ashley Smiley, Kristiina Hurme, Christopher J. Clark and Alejandro Rico-Guevara
DOI: 10.1242/bio.061936

Editor: Lewis Halsey

Review timeline

Original submission:	15 February 2025
Editorial decision:	20 February 2025
First revision received:	24 February 2026
Accepted:	26 February 2026

Original submission

First decision letter

MS ID#: bio.061936

MS Title: Computer vision analysis to identify episodic flapping pauses in hovering hummingbirds

Authors: Maria Ximena Bastidas-Rodriguez; Ana Melisa Fernandes; María José Espejo-Uribe; Diana Abaunza; Juan Sebastián Roncancio; Eduardo Aquiles Gutierrez-Zamora; Cristian Flórez-Pai; Ashley Smiley; Kristiina Hurme; Christopher J. Clark; Alejandro Rico-Guevara

I have now reached a decision on the above manuscript.

The reviewer reports are shown at the bottom of this email or can be accessed, together with a copy of this decision letter, by going to:

As you will see, the reviewers raised a number of substantial criticisms that prevent me from accepting the paper at this stage.

They suggest, however, that a revised version might prove acceptable, if you can address their concerns. If you think that you can deal satisfactorily with the criticisms on revision, I would be pleased to see a revised manuscript. We would then return it to the reviewers.

At this stage, we also ask you to ensure your manuscript complies with our formatting guidelines. Provided you are able to fully address the referees' comments, we are positive about publication of your paper (we accept over 95% of revision submissions) and therefore hope you won't mind any extra work involved in reformatting your manuscript at this point.

Please ensure that you clearly highlight all changes made in the revised manuscript. Please avoid using 'Tracked changes' in Word files as these are lost in PDF conversion.

I should be grateful if you would also provide a point-by-point response detailing how you have dealt with the points raised by the reviewers in the 'Response to Reviewers' box. Please attend to all of the reviewers' comments. If you do not agree with any of their criticisms or suggestions please explain clearly why this is so.

Reviewer 1

Comments for the author

This is a nicely written paper on an interesting topic, and I have no major concerns. The suggestions I list below are designed to make the paper easier for readers to digest.

1. Figures: The images in Figure 2 are too small—I can't see the hummingbird against the background. Also, yellow lines on a white background (figure 2, 3, 4) is difficult to see. The lower panel (there is no panel id shown) of Figure 5 might be better used to show pause frequency manual vs pause frequency algorithm rather than both against body mass because it is difficult to see the discrepancy between the circles and triangles.
2. It would be helpful to indicate for the readers what the typical flap rate for these birds is so that the camera's 500 frames/ second can be understood to be "high-speed." I had to look up hummingbird flap rate (which seems to be ~80 beats/min in flight and as high as 200 beats/min in courtship displays). Do you think that higher speed recordings would decrease the F1-score, FP, or FN?
3. Line 93 of page 1: "...did not have many..." is vague.
4. Line 173 of page 2: "The Mobbs described Coronets..." I'm not sure what this phrase means.
5. Line 204 of Page 2: reference to work in progress is generally not appropriate.

Reviewer 2

Comments for the author

The manuscript presents a novel computational approach for identifying intermittent hovering in hummingbirds using computer vision and signal analysis. The proposed method has the potential to enhance automated detection of wingbeat pauses, therefore contributing to biomechanics and behavioural ecology of birds.

I found it an interesting read and, despite its relatively limited scope, I believe it deserves publication after some major corrections. I have some - mostly methodological - concerns that I think must be addressed to strengthen the validity of the results. Additionally, minor grammatical and formatting errors should be corrected to improve clarity.

Specific comments:

Methodology

- 1- The developed algorithm relies on bounding box similarity to detect pauses. However, slow wingbeats (is this right?) might be misclassified as pauses when consecutive frames are similar. I understand that the first paragraph in Discussion has discussed this issue, but the authors should give a few clear pathways for overcoming this in future research.
- 2- The study acknowledges that illumination conditions were not standardised, which could impact detection accuracy. Could this influence the pause detection? I suggest the authors to test the algorithm's sensitivity to different lighting conditions.
- 3- How many videos were analysed? Method mentions 11 and Results sections says 13.
- 4- Add the number individuals filmed for each species in Methods. Some species represented by only a single individual. Is this current sample size adequate? Is there a possibility for increasing the

number of the analysed videos? If not, I think it would be more appropriate to mention that this study serves as a pilot validation.

5- The number of trained observers has been mentioned in Introduction but not in the methods. Add this info, give inter-rater reliability (IRR) metric, and ensure that the observers were blinded to the algorithm's output during manual annotations.

6- The pause detection threshold is set at 90%. Why is that? Perhaps a sensitivity analysis is required at different thresholds.

7- Can false positive rates be reported for each species separately, as different wingbeat frequencies might affect detection accuracy?

Results

8- Mostly about the validation: the validation is primarily technical and concerns algorithm performance rather than biological, i.e. considering actual kinematic pause verification. Can authors, for example, confirm pause detection with high-speed video motion analysis?

Discussion

9- The discussion assumes that pauses serve energetic, visual, or signalling functions. But what about muscle fatigue or possible neuromuscular constraints?

Minor errors:

10- The name of the species in Fig. 5 should be italic.

11- "Hummingbirds hover, which enables their roles as exclusive pollinators of many plant groups they have co-evolved with." Better to write: "Hummingbirds hover, enabling them to serve as exclusive pollinators for many plant groups with which they have co-evolved."

12- The fonts in most figures are too small that affects readability.

Reviewer's Responses to Questions

Experimental quality

Does each figure have the proper controls?

If 'No', please indicate reasons in Comments for Author box below.

Reviewer #1:

- Yes

Reviewer #2:

- No

Were the data analyzed using appropriate statistical tests?

If 'No', please indicate reasons in Comments for Author box below.

Reviewer #1:

- Yes

Reviewer #2:

- No

Reproducibility

Were experiments performed using adequate number of biological replicates?
If 'No', please indicate reasons in Comments for Author box below.

Reviewer #1:

- Yes

Reviewer #2:

- No

Does the methods section provide sufficient detail to permit reproducibility?
If 'No', please indicate reasons in Comments for Author box below.

Reviewer #1:

- Yes

Reviewer #2:

- No

Completeness

Are the manuscript's conclusions supported by the data?
If 'No', please indicate reasons in Comments for Author box below.

Reviewer #1:

- Yes

Reviewer #2:

- Yes

Scholarship

Do the authors cite and discuss the merits of data that would argue for and against their conclusion?
If 'No', please indicate reasons in Comments for Author box below.

Reviewer #1:

- Yes

Reviewer #2:

- Yes

Does the manuscript title & abstract accurately reflect the contents of the manuscript, without hyperbole?
If 'No', please indicate reasons in Comments for Author box below.

Reviewer #1:

- Yes

Reviewer #2:

- Yes

First revision

Author response to reviewers' comments

First of all, we would like to take advantage of this opportunity to thank the reviewers for their comments and feedback towards improving our manuscript.

Below, we present the modifications, which are based on the recommendations made by the reviewers and are highlighted in the paper with blue font color:

Reviewer #1's comments:

Figures: The images in Figure 2 are too small—I can't see the hummingbird against the background. Also, yellow lines on a white background (figure 2, 3, 4) are difficult to see. The lower panel (there is no panel id shown) of Figure 5 might be better used to show pause frequency manual vs pause frequency algorithm rather than both against body mass because it is difficult to see the discrepancy between the circles and triangles.

Addressed as suggested by the reviewer.

For Figure 5, we modify the text accordingly with the changes made:

Caption:

Pause rate (number of pauses/ number of wingbeats) and pause mean duration, manual vs. algorithm, for 13 individuals of 11 species of hummingbirds Ayerbe-Quiñones (2024).

It would be helpful to indicate for the readers what the typical flap rate for these birds is so that the camera's 500 frames/ second can be understood to be "high-speed." I had to look up hummingbird flap rate (which seems to be ~80 beats/min in flight and as high as 200 beats/min in courtship displays). Do you think that higher speed recordings would decrease the F1-score, FP, or FN?

We added the following to the revised version:

The wingbeat frequency of these species was found to be approximately 35 to 45 Hz (Bastidas-Rodriguez et al., 2024), meaning each wingbeat was captured in about 14 video frames. Videos recorded at higher speeds could augment the amount of frames where a pause/wingbeat is recorded, this information could be very useful for the algorithm as a pause could be easily identified. In turn, this would reduce FN as a pause would last for longer periods of time than other movements, and it would reduce FP as the image quality given by the movement would be smoother, resulting in sharper images on each frame, particularly for the fast-moving wings.

(The comment includes a note about 200 beats/min, but in this context, perhaps it meant 200 beats/second. The suggestion that hummingbirds flap as high as 200 Hz is a myth that has been uncritically repeated by non-authoritative sources; the maximum is ~100 Hz but most species are below 60 Hz)

Line 93 of page 1: "...did not have many..." is vague.

The original phrase:

"[...] and the algorithm did not have many false negatives, [...]" Was changed to:
 "[...]and the algorithm only had 19 false negatives across all videos, [...]"

Line 173 of page 2: "The Mobbs described Coronets..." I'm not sure what this phrase means.

We fixed the citation:

For instance, Mobbs (1982) described Coronets (Genus Boissonneau): [...]

Line 204 of Page 2: reference to work in progress is generally not appropriate.

Removed. Thank you.

Reviewer #2's comments:

The developed algorithm relies on bounding box similarity to detect pauses. However, slow wingbeats (is this right?) might be misclassified as pauses when consecutive frames are similar. I understand that the first paragraph in Discussion has discussed this issue, but the authors should give a few clear pathways for overcoming this in future research.

We added this comment at the end of the Discussion section:

“Further research could use more advanced methods in the field of computer vision, such as landmark detection Lin et al. (2022), to overcome the issues faced here. These methods will probably benefit from using better resolution videos to capture the whole wing movement (e.g., Díaz-Salazar et al. 2024).”

The study acknowledges that illumination conditions were not standardised, which could impact detection accuracy. Could this influence the pause detection? I suggest the authors to test the algorithm's sensitivity to different lighting conditions.

Yes, every computer vision approach would be affected by lighting conditions, and results might vary accordingly. The experiment suggested by the reviewer requires a high amount of data with detected pauses at different illumination levels that are difficult to obtain in the wild (so far, we have not detected the pauses described here in captivity).

How many videos were analysed? Method mentions 11 and Results sections says 13.

We analyzed 13 videos, thank you for catching this discrepancy, we have fixed the typo.

Add the number individuals filmed for each species in Methods. Some species represented by only a single individual. Is this current sample size adequate? Is there a possibility for increasing the number of the analysed videos? If not, I think it would be more appropriate to mention that this study serves as a pilot validation.

We added this comment at the end of the Methods section:

“The proposed method would serve as pilot validation of the pausing behavior detection approach.”

The number of trained observers has been mentioned in Introduction but not in the methods. Add this info, give inter-rater reliability (IRR) metric, and ensure that the observers were blinded to the algorithm's output during manual annotations.

We added the following comment to the Methods section:

“To assess the consistency of the human observer evaluations of both the number and mean duration of pauses, we conducted an intra-rater reliability analysis using the irr package (Gamer et al., 2019). This analysis involved two raters independently evaluating the same set of 13 videos. We employed a two-way random-effects model for the 28 measurements, as recommended by Harvey (2021), where two raters were randomly

selected from a pool of four. The number of pauses and their mean durations, as calculated by the raters, were compared. “

We also added the following analysis to the results section:

“The two raters evaluated the same 14 videos, yielding a single-score intraclass correlation coefficient (ICC) of 0.991 ($p = 1.44e-25$, 95% CI = 0.981 - 0.996). This ICC value exceeds the 0.80 threshold for substantial agreement set by Gwet (2014), indicating an almost perfect correlation as defined by Cohen (1968) and Harvey (2021).”

The pause detection threshold is set at 90%. Why is that? Perhaps a sensitivity analysis is required at different thresholds.

Yes, it is true, due to the small number of samples this was the number that best reduced the amount of FP. But it would vary with a larger amount of data. The sensitivity analysis suggested is out of the scope of the current paper.

Can false positive rates be reported for each species separately, as different wingbeat frequencies might affect detection accuracy?

These are documented in the supplementary material document.

Mostly about the validation: the validation is primarily technical and concerns algorithm performance rather than biological, i.e. considering actual kinematic pause verification. Can authors, for example, confirm pause detection with high-speed video motion analysis?

We decided to perform the analysis presented in the paper by using the validation of trained human observers. We believe that the ground truth given by them is enough to confirm the pause detection on the analyzed videos.

We also added the IRR analysis kindly suggested to obtain the reliability score of the presented method.

The discussion assumes that pauses serve energetic, visual, or signalling functions. But what about muscle fatigue or possible neuromuscular constraints?

We have added the following to the discussion:

“Additional hypotheses around intermittent hovering are worth exploring in future work, for instance, its relationship with muscle fatigue or possible neuromuscular constraints (Mahalingam and Welch 2013, Tobalske 2010).”

The name of the species in Fig. 5 should be italic.

Addressed as suggested, thank you!

“Hummingbirds hover, which enables their roles as exclusive pollinators of many plant groups they have co-evolved with.” Better to write: “Hummingbirds hover, enabling them to serve as exclusive pollinators for many plant groups with which they have co-evolved.”

Addressed as suggested, thank you!

The fonts in most figures are too small that affects readability.

Addressed accordingly, thanks!

Second decision letter

MS ID#: bio.061936R1

MS Title: Computer vision analysis to identify episodic flapping pauses in hovering hummingbirds

Authors: Maria Ximena Bastidas-Rodriguez; Ana Melisa Fernandes; María José Espejo-Uribe; Diana Abaunza; Juan Sebastián Roncancio; Eduardo Aquiles Gutierrez-Zamora; Cristian Flórez-Pai; Ashley Smiley; Kristiina Hurme; Christopher J. Clark; Alejandro Rico-Guevara

This afternoon, I have read through your responses to the reviewers along with your tracked changes to the manuscript. I am happy to tell you that your manuscript has been accepted for publication in Biology Open, pending our standard publication integrity checks. It was accepted on 26th February 2026.